# Body Mass Index and Weight Change as Predictors of Hypertension Development: A Sex-Specific Analysis

**DOI:** 10.3390/nu17010119

**Published:** 2024-12-30

**Authors:** Koki Kosami, Masanari Kuwabara, Akira Okayama, Ryusuke Ae

**Affiliations:** 1Division of Public Health, Center for Community Medicine, Jichi Medical University, 3311-1 Yakushiji, Shimotsuke 329-0498, Japan; k.kosami@jichi.ac.jp (K.K.); shirouae@jichi.ac.jp (R.A.); 2Division of Cardiovascular Medicine, Department of Medicine, Jichi Medical University, Shimotsuke 329-0498, Japan; 3Research Institute of Strategy for Prevention, Tokyo 103-0006, Japan; aokayama55@gmail.com

**Keywords:** obesity, hypertension, body mass index, weight change, epidemiology

## Abstract

**Background**/**Objectives**: Obesity is associated with food intake, drinking, and nutrition. It is necessary to examine the association between body mass index (BMI), body weight changes, and the incidence of hypertension, focusing on potential sex differences. **Methods**: A retrospective two-point cohort study was conducted using data from the Japanese Specific Health Checkup program from 2011 to 2013. Multivariable logistic regression analyses were employed to assess associations between BMI, weight change, and hypertension onset, adjusting for age, sex, and lifestyle factors, including smoking, drinking, and exercise. **Results**: In total, 397,181 participants were analyzed. A higher BMI was linked to a higher risk of hypertension, with an odds ratio (OR) of 1.11 (95% confidence interval [CI]: 1.11–1.12) for both sexes. Women aged 40–65 showed higher ORs than men. Weight gain was associated with developing hypertension, with an OR (95% CI) of 1.09 (1.09–1.10) in men and 1.08 (1.07–1.08) in women. This association held across all BMI levels, even among individuals without obesity, with the effect being generally stronger in men. **Conclusions**: Both BMI and weight change contribute to the development of hypertension among the Japanese healthy population, with differences based on sex. Weight reduction may reduce the risk of hypertension for individuals with and without obesity, emphasizing the importance of weight stability through food and nutrition control, particularly for women.

## 1. Introduction

Obesity is associated with physical inactivity and excessive calorie intake, especially high intakes of fructose, which has been linked to increased fat accumulation in the liver and insulin resistance [1]. Obesity represents a global health issue because the prevalence of obesity has increased more than three times in men and more than twice in women in the past four decades [2,3]. The World Health Organization (WHO) has reported that over 4 billion people may be overweight or obese compared with approximately 2.6 billion in 2020 [4]. A high body mass index (BMI) is one of the major risk factors for the global disease burden, particularly in developed countries. BMI interacts with metabolic risk factors such as high blood glucose, high blood pressure, low physical activity, and the consumption of sugar-sweetened beverages, necessitating interventions focused on obesity and metabolic syndrome [5]. Obesity increases the risk of comorbidities such as hypertension, diabetes, and dyslipidemia, which can result in cardiovascular disease [6,7,8,9,10,11,12]. However, body weight reduction improves comorbidities and may reduce systolic and diastolic blood pressure [6,12,13,14,15,16].

The WHO defines overweight as a BMI of 25 kilograms (kg)/meter (m)^2^ or higher and below 30 kg/m^2^, and obesity as a BMI of 30 kg/m^2^ or higher [4]. However, the criteria for the Japanese population differ from the WHO criteria: the Japan Society for the Study of Obesity (JASSO) defines obesity as a BMI of 25 kg/m^2^ or higher because the BMI associated with the lowest morbidity in the Japanese population is approximately 22 kg/m^2^ [17], and the proportion of individuals with obesity (BMI of 30 or higher) is low in Japan [18]. This suggests that Asian populations may be more sensitive to body weight changes than Western populations [19,20,21,22,23]. A study showed that the Japanese BMI cut-off that is associated with cardiovascular risk factors is significantly lower than the American BMI cut-off [24]. Despite the well-known association between obesity and various comorbidities, knowledge regarding BMI effects and body weight changes in both populations with and without obesity in Japan remains insufficient. We conducted a retrospective cohort study using an existing health check database of the Japanese healthy population to investigate the effects of BMI and body weight changes on the incidence of comorbidities, with a particular focus on hypertension.

## 2. Methods

### 2.1. Study Design and Study Participants

This is a retrospective two-point cohort study using data from the Specific Health Checkup (SHC) system, which is a nationwide annual health screening program specifically targeting metabolic syndrome [25,26]. In this study, the 2011 checkup served as the baseline, with BMI and body weight changes over the two-year period (2011–2013) evaluated as predictors of hypertension development in 2013, making this a two-point cohort study. We investigated the association between the onset of hypertension and BMI and body weight change from 2011 to 2013.

In Japan, it is mandatory that all residents enroll in medical insurance based on their age and occupation. Individuals younger than 75 years old must enroll in either the Employee’s Health Insurance, which covers employees and their dependents, or the National Health Insurance, which covers self-employed individuals, retirees, and their dependents. Local governments provide SHCs annually for insured persons aged 40–75 under these two insurance systems. The SHC includes checks on medical history, lifestyle, blood pressure, and various biomarkers. We obtained anonymized SHC records from 155 local health insurance organizations in Japan.

This study was approved by the Ethics Review Board of Jichi Medical University (Approval ID: 22-232, 25 September 2023).

### 2.2. Inclusion and Exclusion Criteria

The inclusion criteria were as follows: having a record of SHC in the 155 local health insurance organizations and having records from both 2011 and 2013. A total of 1,070,268 individuals were screened for this study. Exclusion criteria were as follows: participants who lacked information on blood pressure, drinking habits, exercise habits, or smoking habits; those with pre-existing hypertension in 2011 (the definition of hypertension is described in Section 2.3); and participants with extreme values for body weight (those with a BMI less than 18.5 kg/m^2^, a BMI greater than 40 kg/m^2^, or a body weight change of more than 10 kg).

### 2.3. Measurements

The primary outcome was the onset of hypertension. Hypertension was defined as meeting any of the following three conditions: systolic blood pressure of 140 mmHg or higher, diastolic blood pressure of 90 mmHg or higher, or taking medication for hypertension. We considered a participant to have developed hypertension if they met the definition of hypertension. The exposure variables were BMI in 2011 and body weight change from 2011 to 2013.

BMI was calculated by dividing body weight (kg) by the square of body height (m) using the SHC records from 2011 and was stratified into six groups: 18.5 ≤ BMI < 20; 20 ≤ BMI < 22; 22 ≤ BMI < 25; 25 ≤ BMI < 30; 30 ≤ BMI < 35; and 35 ≤ BMI < 40. In this study, obesity was defined as a BMI of 25 or higher, based on the JASSO’s criteria. Body weight change was defined as the difference between body weight in 2011 and 2013 and was stratified into seven groups: <−2.5; −2.5≤ and <−1.5; −1.5≤ and <−0.5; −0.5≤ and <0.5; 0.5≤ and <1.5; 1.5≤ and <2.5; and 2.5≤. The lifestyle variables—drinking habits [27,28], exercise habits [29,30], and smoking habits [31,32]—were extracted from the database as confounders. These variables were treated as dichotomous and represented as either presence or absence.

In the additional sensitivity analyses, we adjusted for diabetes (defined as HbA1c ≥ 6.5%) and dyslipidemia (LDL cholesterol ≥ 140 mg/dL, HDL cholesterol < 40 mg/dL, or triglycerides ≥ 150 mg/dL) according to the International Expert Committee and the Japan Atherosclerosis Society guidelines [33,34].

### 2.4. Statistical Analysis

The cumulative incidences of hypertension were calculated for each BMI and body weight change group. The number of participants who developed hypertension was divided by the total number of participants within each BMI and body weight change category. A multivariable logistic regression analysis was used to calculate the adjusted odds ratios (ORs) and confidence intervals (CIs) of the exposure variables for the onset of hypertension, adjusting for relevant lifestyle factors. The onset of hypertension was treated as the dependent variable, while lifestyle factors—drinking habits, exercise habits, and smoking habits—were included as independent variables. All analyses were stratified by sex and age groups.

Four types of exposures were incorporated into the model. The first analysis estimated the adjusted ORs of BMI in 2011 for the onset of hypertension. The model included BMI in 2011, age, and lifestyle factors as independent variables. The second analysis estimated the adjusted ORs of body weight change. This model included body weight change, BMI in 2011, age, and lifestyle factors as independent variables. The third analysis estimated the adjusted ORs of body weight change stratified by the presence of obesity. The model included body weight change and lifestyle factors and was stratified into three groups: BMI ranges of 18.5 ≤ BMI < 25; 25 ≤ BMI < 30; and 30 ≤ BMI < 40. The fourth analysis estimated the adjusted ORs of body weight change in participants with obesity and those without obesity. The model included body weight change and lifestyle factors and was stratified into three groups: BMI ranges of 18.5≤ and <20; 20≤ and <22; and 22≤ and <25.

All statistical analyses were performed using R version 4.4.0.

## 3. Results

### 3.1. Participants

A total of 1,070,268 participants had records from both 2011 and 2013. Initially, 275,175 participants were excluded because of missing data (146,600 lacked blood pressure, 70,262 lacked drinking habits, 58,195 lacked exercise habits, and 118 lacked smoking habits). Additionally, 397,912 participants did not meet the eligibility criteria for analysis (353,834 had hypertension in 2011, 41,622 had a BMI under 18.5, 56 had a BMI over 40, 6 lacked BMI, and 2394 experienced a body weight change of over 10 kg). Finally, 397,181 participants were included in the analysis (Figure 1). The participants included 243,825 women (61.4%) and 153,356 men (38.6%), with a mean age (standard deviation, SD) of 62 (8) years. The mean (SD) BMI in 2011 was 22.58 (2.61) kg/m^2^, and 64,617 participants (17%) had a BMI over 25.0 kg/m^2^ in 2011. The mean (SD) systolic blood pressure was 120 (12) mmHg, and the mean diastolic blood pressure was 72 (9) mmHg. Regarding lifestyle factors, 50,067 participants (13%) had a smoking habit, 182,490 participants (46%) had a drinking habit, and 253,146 participants (64%) had an exercise habit. The mean (SD) body weight change over 2 years was −0.23 (2.31) kg, and 84,079 participants (21%) experienced no change in body weight (Table 1). The basic characteristics by sex are presented in Table 2 (men) and Table 3 (women).

### 3.2. Cumulative Incidence of Hypertension

In 2013, 59,777 participants (15%) developed hypertension, including 33,303 women (14%) and 26,474 men (17%). The distribution of the cumulative incidence rate by BMI in 2011 and body weight change is shown in Figure 2. Compared with the group with no body weight change (−0.5≤ and <0.5 group), weight gain was associated with an increased cumulative incidence rate of hypertension across all BMI groups. Conversely, weight loss did not significantly reduce the cumulative incidence rate of hypertension in certain BMI groups (30 ≤ BMI < 35, and 35 ≤ BMI < 40 groups). The highest cumulative incidence rate (35.5%) was observed in the group with a BMI range of 35 ≤ BMI < 40 and a weight change range of 2.5≤. The lowest incidence rate (8.7%) was observed in the group with a BMI range of 18.5≤ and <20 and a weight change range of −2.5≤ and <−1.5.

### 3.3. Association Between the Incidence of Hypertension and BMI and Weight Change

An increase of one point in BMI was associated with the incidence of hypertension, with an OR (95% CI) of 1.11 (1.11, 1.12) in men and 1.11 (1.11, 1.12) in women. When stratified by age groups, the adjusted ORs in women were higher than those in men in the 40≤ and <55, and 55≤ and <65 age groups (Figure 3a). Similarly, an increase of one-kg in body weight was associated with the incidence of hypertension, with an adjusted OR (95% CI) of 1.09 (1.09, 1.10) in men and 1.08 (1.07, 1.08) in women. However, the ORs in women were lower than those in men across all age groups (Figure 3b).

When analyses of body weight change were stratified by BMI, an increase of one-kg in body weight was associated with the incidence of hypertension across all BMI categories: the non-obese group (18.5 ≤ BMI < 25), the obese group (25 ≤ BMI < 30), and the severely obese group (30 ≤ BMI < 40). These associations were observed across all age groups (Figure 4a–c). In the non-obese group, body weight change was associated with the incidence of hypertension in all BMI subgroups. Among the BMI groups of 20 ≤ BMI < 22 and 22 ≤ BMI < 25, the ORs in women were lower than those in men across all age groups. However, this trend was not observed in the BMI group of 18.5 ≤ BMI < 20 (Figure 4d–f).

## 4. Discussion

This study revealed two key findings. First, both a higher baseline BMI and increased body weight change were associated with the incidence of hypertension, and these effects differed between men and women. The effect of BMI was more pronounced in women, particularly among younger participants, whereas the effect of body weight change was stronger in men. Second, an increased body weight change was associated with the incidence of hypertension even in participants without obesity, and the effect was more significant in men.

The associations of both BMI and body weight changes with hypertension were consistent with the findings from several previous studies [8,9]. Additionally, our study indicated a clear difference by sex. The effect of BMI was greater in women, particularly in younger age groups. Referring to the basic characteristics by sex (Table 1, Table 2 and Table 3), we observed that the BMI distribution also differed by sex, with female participants having a lower BMI compared with male participants. This trend is consistent with Japanese national statistics [18,35], suggesting that a higher BMI is a less common condition among women, which could predispose women to hypertension. In contrast, the effect of body weight change appeared to be smaller in women, possibly because of differences in the BMI distribution. For the same degree of body weight change, the impact on the development of hypertension may be less significant in women compared to men. Therefore, maintaining a stable weight may be particularly important for women. Moreover, our results indicated that body weight change was associated with the development of hypertension not only in participants with obesity, but also in participants without obesity. Even among those with a BMI below 25 kg/m^2^, increased body weight change was related to the development of hypertension. However, this finding does not suggest that it should be recommended that individuals with a low BMI lose weight, as this study focused only on hypertension. Recommendations for weight reduction should consider overall health status. Nonetheless, modest weight reduction could be beneficial for individuals who are not underweight.

Our results indicated that the effects of BMI and body weight change on the development of hypertension differ by sex. Two factors are considered to explain this difference: the distribution of obesity by sex and the impact of menopause. First, previous studies in Japan have reported that the proportion of obesity is lower in women, especially in younger populations [18]. Additionally, polarization between Japanese women with obesity and those without obesity has been reported, with individuals without obesity tending to lose weight, while individuals with obesity tend to gain weight [36]. This trend suggests that Japanese women with obesity may pay less attention to their body weight compared to women without obesity, potentially contributing to a higher risk of hypertension. Second, pregnancy and menopause are associated with the development of several comorbidities, such as obesity and hypertension, through changes in sex hormones [37,38,39]. Most of the female participants in this study had likely reached menopause, as the median age of female participants was 63 years, and only individuals aged 40 or older were included. Therefore, in postmenopausal women, the effect of BMI could be greater, and the effect of body weight change smaller, compared to male participants due to the predisposition to hypertension associated with hormonal change after menopause. In other words, maintaining a stable body weight might be important for women before menopause to prevent hypertension. In fact, sex differences are well documented, and recent research has directly assessed hormonal factors related to menopause [40,41]. These studies provide substantial evidence for biological explanations of biological mechanisms underlying health changes in postmenopausal women, including increased cardiovascular risk, bone density loss, and metabolic alterations. By integrating these findings, our study aligns with the broader body of evidence supporting hormonal influences on sex-specific health outcomes. Recently, salt intake and umami-rich foods have been shown to induce obesity, primarily through increased calorie intake [42,43]. Salt intake is a well-known factor associated with elevated blood pressure. Although this study could not assess detailed food intake information, participants with increased body weight might consume higher amount of salt, umami-rich foods, and fructose, which could contribute to the development of hypertension. Therefore, further studies assessing food intake and nutrition are needed to better understand the cause of obesity and weight gain.

This study has several limitations. First, the participants of this study may not be representative of the general Japanese population because the inclusion criteria required having health check records from both 2011 and 2013, as well as no missing data. As a result, the participants in this study may possess better health literacy compared with the general population. However, even if the study participants were healthier than the general population, the results would likely result in an underestimation of the association, leaving our interpretation largely unaffected. Second, this study has a selection bias because our data were limited to 2011 and 2013 due to data availability. Moreover, this study cannot show causality between BMI/weight changes and the development of hypertension due to its retrospective observational design. Third, this study did not account for factors that may have contributed to body weight changes over the 2 years. Potential confounders such as dietary improvements or increased physical activity levels may have contributed to the observed differences across the age groups and sexes. Additionally, this study lacks some important information, such as diet, socioeconomic status, and genetic predispositions, due to the nature of a retrospective database study. Therefore, further studies are needed to incorporate this information. Fourth, hypertension may not be a sufficient outcome for this study. Although obesity is associated with other comorbidities such as diabetes and dyslipidemia, this study focused on hypertension as the main outcome of obesity. To address this limitation, we conducted additional sensitivity analyses that included diabetes and dyslipidemia as covariates. The results, shown in Figure 5 and Figure 6, were similar to the main results described above. Fifth, we created and used categories of BMI and body weight change in this study. However, this method might oversimplify complex relationships, particularly for individuals near category thresholds. Moreover, this study assessed only a two-year interval and, therefore, it may not capture the long-term effects of BMI and weight changes on the development of hypertension. Sixth, we relied on odds ratios for the entire population in this study, which may obscure subgroup differences, particularly in smaller BMI categories. To address this, we conducted subgroup analyses by BMI categories, which yielded consistent results. However, variations may still exist due to the limited sample sizes, and further studies are needed to explore these differences. Seventh, although potential environmental or cultural contributors, such as stress levels, are important factors for hypertension, we could not assess them due to data limitations. This remains a limitation, and future studies incorporating these factors are needed for a more comprehensive understanding of hypertension risk. Finally, our results are difficult to generalize to people in other countries because we focused specifically on a Japanese population. The data were collected through the SHC system, a national health checkup program that mandates participation for all Japanese residents over the age of 40. Because of the potential differences in BMI distributions and health conditions between Japan and other regions, such as East Asia or Western countries, the direct applicability of these findings may be limited. Therefore, caution should be exercised when interpreting the results in the context of other populations.

### Strengths and Weaknesses of This Study

This study’s strengths include a large sample size of 397,181 participants, providing robust statistical power, and the use of standardized data from the Japanese Specific Health Checkup program. The analysis of sex differences offers valuable insights into hypertension risk factors for men and women.

However, the two-year interval may not capture long-term effects, and factors such as diet, socioeconomic status, and stress levels were not evaluated due to data limitations. Additionally, categorizing BMI and weight changes may oversimplify complex relationships, particularly near category thresholds. Furthermore, as a retrospective cohort study, it cannot establish causal relationships.

## 5. Conclusions

This study indicated an association between the development of hypertension and both BMI and body weight change in the Japanese population. This association differed by sex. Weight reduction could potentially help to prevent hypertension not only in individuals with obesity, but also in those without obesity. Moreover, maintaining a stable weight could be especially important for women in managing hypertension risk.

## Figures and Tables

**Figure 1 nutrients-17-00119-f001:**
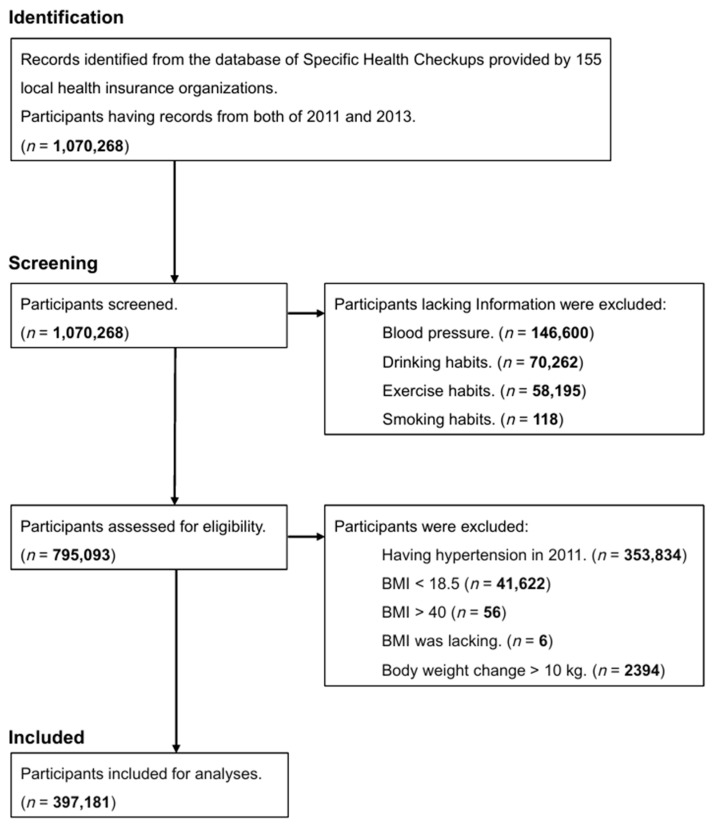
Study participants. Abbreviations: *n*, number of participants, kg, kilogram; BMI, body mass index (kg/m^2^).

**Figure 2 nutrients-17-00119-f002:**
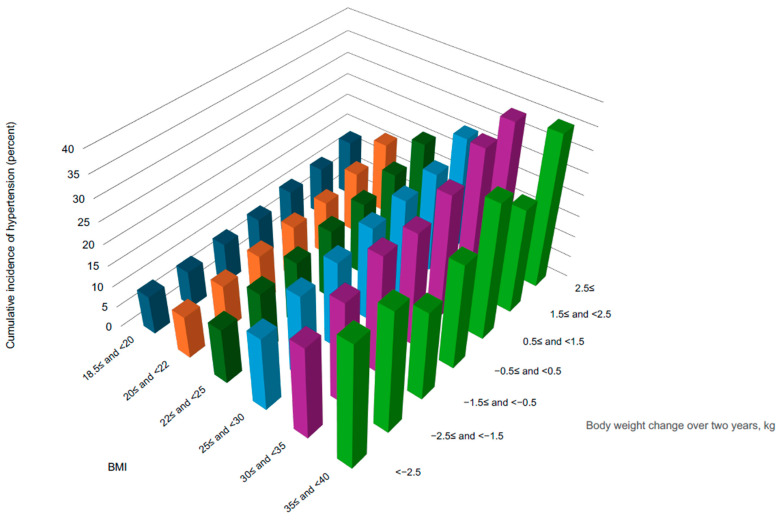
Cumulative incidence rate of hypertension by body mass index (kg/m^2^) and body weight change (kg). Abbreviations: kg, kilogram; BMI, body mass index (kg/m^2^).

**Figure 3 nutrients-17-00119-f003:**
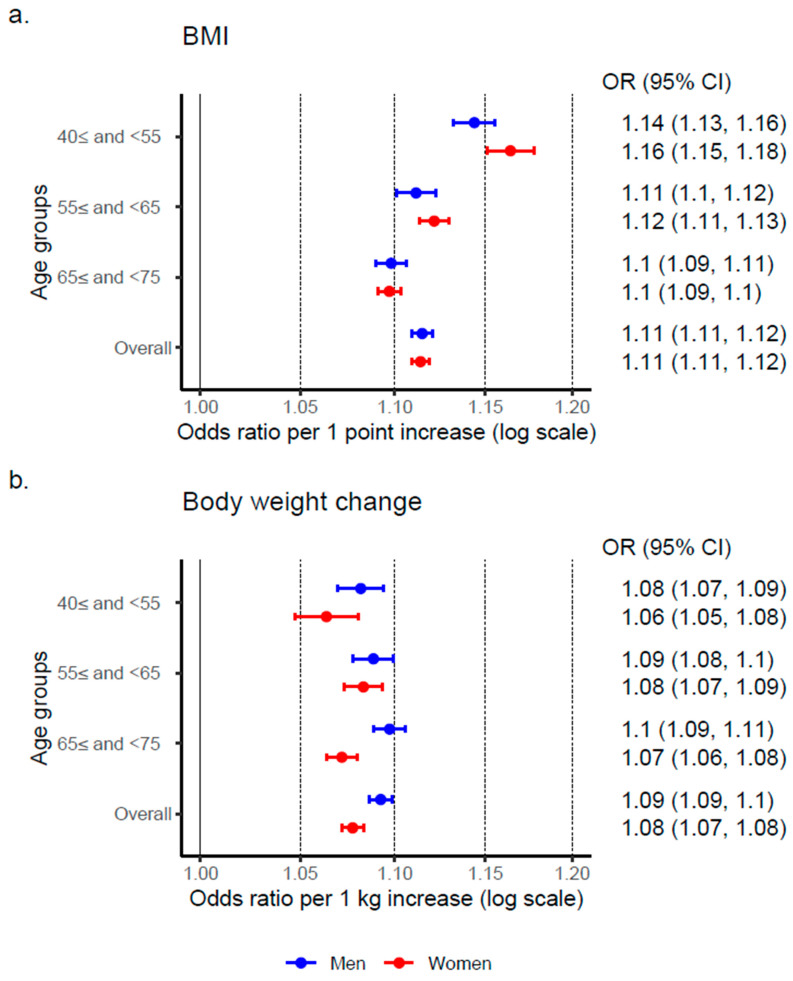
Association between the incidence of hypertension and body mass index and body weight change. Abbreviations: kg, kilogram; BMI, body mass index (kg/m^2^); OR, odds ratio; CI, confidence interval. (**a**) The odds ratios of BMI in 2011 for the onset of hypertension were calculated for each stratified age group. The model included BMI in 2011, age, and lifestyle factors as independent variables. (**b**) The odds ratios of body weight change for the onset of hypertension were calculated for each stratified age group. The model included body weight change, BMI in 2011, age, and lifestyle factors as independent variables.

**Figure 4 nutrients-17-00119-f004:**
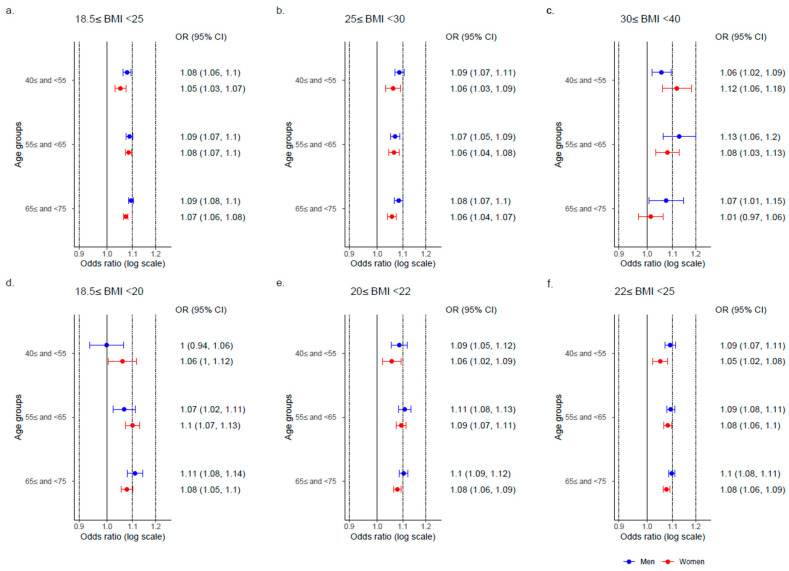
Association between hypertension onset and body weight change across body mass index groups. Abbreviations: BMI, body mass index (kg/m^2^); OR, odds ratio; CI, confidence interval. The odds ratios represent the value per one-kilogram increase. The odds ratios of body weight change stratified by BMI in 2011 were calculated. The model included body weight change and lifestyle factors.

**Figure 5 nutrients-17-00119-f005:**
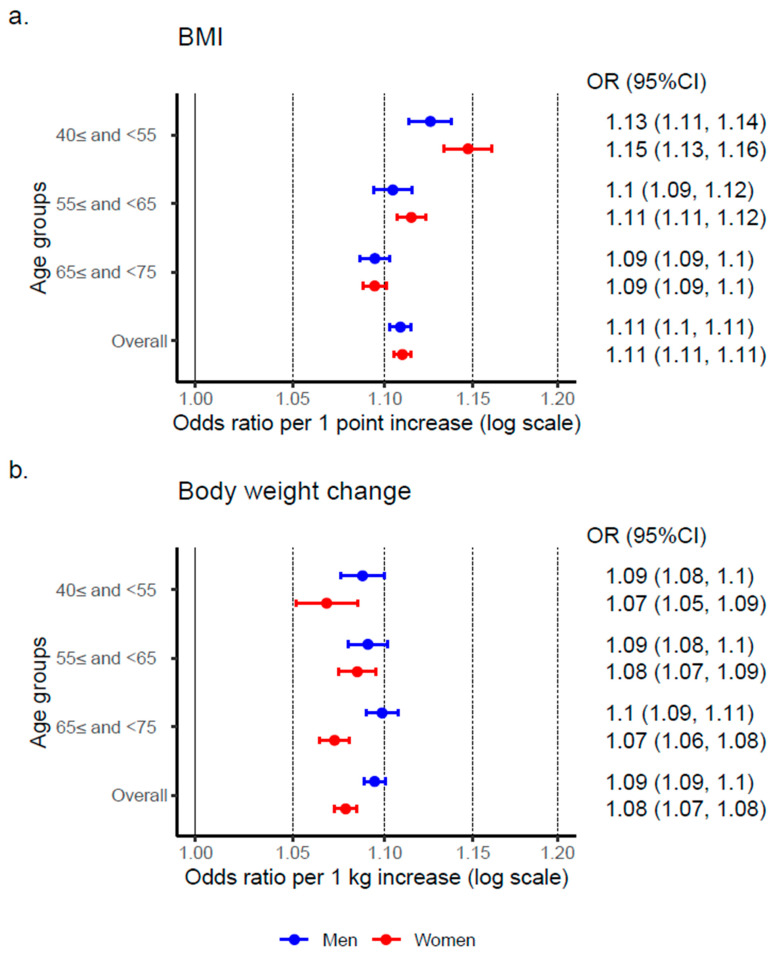
Association between the incidence of hypertension and body mass index and body weight change based on sensitivity analyses incorporating diabetes and dyslipidemia. Abbreviations: BMI, body mass index; OR, odds ratio; CI, confidence interval. (**a**) The odds ratios of BMI in 2011 for the onset of hypertension were calculated for each stratified age group. The model included BMI in 2011, age, diabetes, dyslipidemia, and lifestyle factors as independent variables. (**b**) The odds ratios of body weight change for the onset of hypertension were calculated for each stratified age group. The model included body weight change, BMI in 2011, age, diabetes, dyslipidemia, and lifestyle factors as independent variables.

**Figure 6 nutrients-17-00119-f006:**
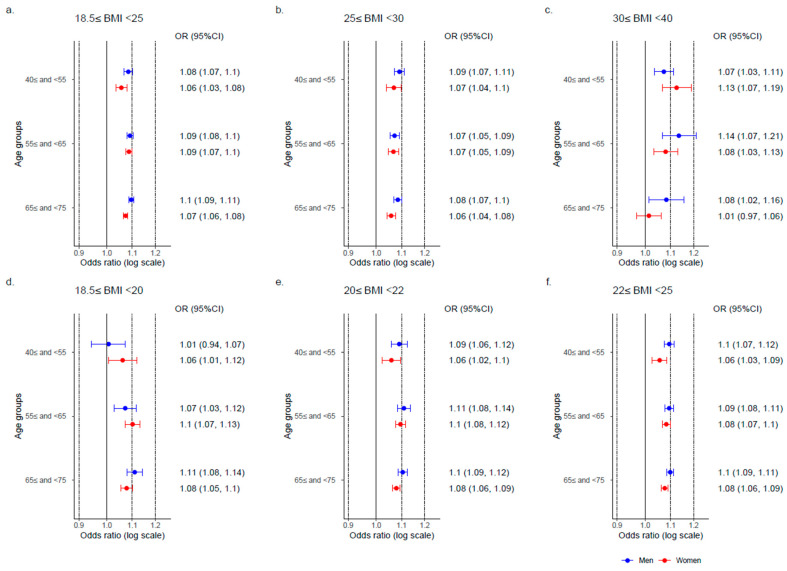
Association between hypertension onset and body weight change across body mass index groups based on sensitivity analyses incorporating diabetes and dyslipidemia. Abbreviations: BMI, body mass index (kg/m^2^); OR, odds ratio; CI, confidence interval. The odds ratios represent the value per one-kilogram increase. The odds ratios of body weight change stratified by BMI in 2011 were calculated. The model included body weight change, diabetes, dyslipidemia, and lifestyle factors.

**Table 1 nutrients-17-00119-t001:** Basic characteristics of participants.

Characteristic	Overall, *n* = 397,181	Women, *n* = 243,825	Men, *n* = 153,356
Age, mean (SD), years	62 (8)	63 (8)	62 (9)
Age group, *n* (%)			
≥40 and <55 years old	69,677 (18%)	36,025 (15%)	33,652 (22%)
≥55 and <65 years old	133,442 (34%)	90,663 (37%)	42,779 (28%)
≥65 and <75 years old	194,062 (49%)	117,137 (48%)	76,925 (50%)
BMI, mean (SD)	22.58 (2.61)	22.21 (2.57)	23.16 (2.56)
BMI group, *n* (%)			
18.5 ≤ BMI < 20 kg/m^2^	63,358 (16%)	48,910 (20%)	14,448 (9.4%)
20 ≤ BMI < 22 kg/m^2^	120,854 (30%)	81,731 (34%)	39,123 (26%)
22 ≤ BMI < 25 kg/m^2^	148,352 (37%)	80,966 (33%)	67,386 (44%)
25 ≤ BMI < 30 kg/m^2^	59,689 (15%)	29,386 (12%)	30,303 (20%)
30 ≤ BMI < 35 kg/m^2^	4540 (1.1%)	2600 (1.1%)	1940 (1.3%)
35 ≤ BMI < 40 kg/m^2^	388 (<0.1%)	232 (<0.1%)	156 (0.1%)
Systolic blood pressure, mean (SD), mmHg	120 (12)	120 (12)	121 (11)
Diastolic blood pressure, mean (SD), mmHg	72 (9)	71 (9)	74 (8)
Smoking habit, *n* (%)	50,067 (13%)	12,739 (5.2%)	37,328 (24%)
Drinking habit, *n* (%)	182,490 (46%)	79,327 (33%)	103,163 (67%)
Exercise habit, *n* (%)	253,146 (64%)	154,173 (63%)	98,973 (65%)
Body weight change, mean (SD), kg	−0.23 (2.31)	−0.24 (2.17)	−0.22 (2.53)
Body weight change group, No. (%)			
<−2.5 kg	52,112 (13%)	29,036 (12%)	23,076 (15%)
−2.5≤ and <−1.5 kg	44,251 (11%)	26,847 (11%)	17,404 (11%)
−1.5≤ and <−0.5 kg	69,654 (18%)	44,498 (18%)	25,156 (16%)
−0.5≤ and <0.5 kg	84,079 (21%)	55,277 (23%)	28,802 (19%)
0.5≤ and <1.5 kg	67,772 (17%)	43,321 (18%)	24,451 (16%)
1.5≤ and <2.5 kg	39,723 (10%)	23,971 (9.8%)	15,752 (10%)
2.5 kg≤	39,590 (10.0%)	20,875 (8.6%)	18,715 (12%)
Dyslipidemia, *n* (%)	217,790 (55%)	137,543 (56%)	80,247 (52%)
Diabetes, *n* (%)	25,405 (6.4%)	10,581 (4.3%)	14,824 (9.7%)

Abbreviations: *n*, number of participants, SD, standard deviation; BMI, body mass index. Body weight change was the difference in the body weights in 2011 and 2013.

**Table 2 nutrients-17-00119-t002:** Basic characteristics of male participants by age group.

Characteristic	40 ≤ Age < 55, *n* = 33,652	55 ≤ Age < 65, *n* = 42,779	65 ≤ Age < 75, *n* = 76,925
Age, mean (SD), years	47 (4)	61 (3)	69 (2)
BMI, mean (SD)	23.66 (2.97)	23.21 (2.53)	22.92 (2.34)
BMI group, *n* (%)			
≥18.5 and <20 kg/m^2^	2817 (8.4%)	3920 (9.2%)	7711 (10%)
≥20 and <22 kg/m^2^	7774 (23%)	10,653 (25%)	20,696 (27%)
≥22 and <25 kg/m^2^	13,603 (40%)	18,744 (44%)	35,039 (46%)
≥25 and <30 kg/m^2^	8354 (25%)	8963 (21%)	12,986 (17%)
≥30 and <35 kg/m^2^	990 (2.9%)	475 (1.1%)	475 (0.6%)
≥35 and <40 kg/m^2^	114 (0.3%)	24 (<0.1%)	18 (<0.1%)
Systolic blood pressure, mean (SD), mmHg	118 (11)	122 (11)	123 (11)
Diastolic blood pressure, mean (SD), mmHg	74 (9)	75 (8)	74 (8)
Smoking habit, *n* (%)	11,882 (35%)	11,689 (27%)	13,757 (18%)
Drinking habit, *n* (%)	23,057 (69%)	29,183 (68%)	50,923 (66%)
Exercise habit, *n* (%)	17,314 (51%)	25,197 (59%)	56,462 (73%)
Body weight change, mean (SD), kg	−0.02 (2.94)	−0.25 (2.54)	−0.30 (2.31)
Body weight change group, *n* (%)			
<−2.5 kg	5639 (17%)	6559 (15%)	10,878 (14%)
≥−2.5 and <−1.5 kg	3316 (9.9%)	4955 (12%)	9133 (12%)
≥−1.5 and <−0.5 kg	4637 (14%)	6996 (16%)	13,523 (18%)
≥−0.5 and <0.5 kg	5322 (16%)	7921 (19%)	15,559 (20%)
≥0.5 and <1.5 kg	5090 (15%)	6723 (16%)	12,638 (16%)
≥1.5 and <2.5 kg	3777 (11%)	4396 (10%)	7579 (9.9%)
≥2.5 kg	5871 (17%)	5229 (12%)	7615 (9.9%)
Dyslipidemia, *n* (%)	17,890 (53%)	23,321 (55%)	39,036 (51%)
Diabetes, *n* (%)	1464 (4.4%)	4201 (9.8%)	9159 (12%)

Abbreviations: *n*, number of participants, SD, standard deviation; BMI, body mass index.

**Table 3 nutrients-17-00119-t003:** Basic characteristics of female participants by age group.

Characteristic	40 ≤ Age < 55, *n* = 36,025	55 ≤ Age < 65, *n* = 90,663	65 ≤ Age < 75, *n* = 117,137
Age, mean (SD), years	47 (4)	61 (3)	69 (2)
BMI, mean (SD)	22.09 (2.88)	22.18 (2.55)	22.28 (2.48)
BMI group, *n* (%)			
≥18.5 and <20 kg/m^2^	9061 (25%)	18,461 (20%)	21,388 (18%)
≥20 and <22 kg/m^2^	12,173 (34%)	30,676 (34%)	38,882 (33%)
≥22 and <25 kg/m^2^	9792 (27%)	29,820 (33%)	41,354 (35%)
≥25 and <30 kg/m^2^	4228 (12%)	10,705 (12%)	14,453 (12%)
≥30 and <35 kg/m^2^	660 (1.8%)	938 (1.0%)	1002 (0.9%)
≥35 and <40 kg/m^2^	111 (0.3%)	63 (<0.1%)	58 (<0.1%)
Systolic blood pressure, mean (SD), mmHg	113 (13)	119 (12)	122 (12)
Diastolic blood pressure, mean (SD), mmHg	69 (9)	72 (9)	72 (9)
Smoking habit, *n* (%)	4640 (13%)	4827 (5.3%)	3272 (2.8%)
Drinking habit, *n* (%)	16,313 (45%)	30,472 (34%)	32,542 (28%)
Exercise habit, *n* (%)	18,242 (51%)	53,619 (59%)	82,312 (70%)
Body weight change, mean (SD), kg	0.00 (2.53)	−0.23 (2.16)	−0.32 (2.05)
Body weight change group, No. (%)			
<−2.5 kg	4741 (13%)	10,737 (12%)	13,558 (12%)
≥−2.5 and <−1.5 kg	3399 (9.4%)	9894 (11%)	13,554 (12%)
≥−1.5 and <−0.5 kg	5507 (15%)	16,554 (18%)	22,437 (19%)
≥−0.5 and <0.5 kg	7064 (20%)	20,497 (23%)	27,716 (24%)
≥0.5 and <1.5 kg	6204 (17%)	16,283 (18%)	20,834 (18%)
≥1.5 and <2.5 kg	4104 (11%)	8956 (9.9%)	10,911 (9.3%)
≥2.5 kg	5006 (14%)	7742 (8.5%)	8127 (6.9%)
Dyslipidemia, *n* (%)	11,230 (31%)	53,560 (59%)	72,753 (62%)
Diabetes, *n* (%)	533 (1.5%)	3717 (4.1%)	6331 (5.4%)

Abbreviations: *n*, number of participants, SD, standard deviation; BMI, body mass index.

## Data Availability

The original contributions presented in this study are included in the article. Further inquiries can be directed to the first author (K.K.). K.K. had full access to all of the data and takes responsibility for the integrity of the data and the accuracy of the data analysis. The data are not publicly available due to privacy, legal, or ethical reasons.

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
