# Peer review of "Body Mass Index and Weight Change as Predictors of Hypertension Development: A Sex-Specific Analysis"

_nutrients, 2024, doi:10.3390/nu17010119_

Round 1
Reviewer 1 Report
Comments and Suggestions for Authors
Dear ,
The manuscript by Kosami et al. evaluates the associations between BMI, body weight changes, and the risk of hypertension using data from a Japanese population. While the study addresses a topic of potential significance, I must highlight several methodological weaknesses and issues in the presentation of results that hinder the ability to draw clear and reliable conclusions.
1. Study Design:
The authors claim this is a retrospective cohort study. However, key elements of a cohort study, such as the index date, data collection period, and follow-up duration, are not clearly presented. For instance, the body weight data were collected in 2011 and 2013, but the personal interval between these two years may vary significantly (between 366 and 731 days). Additionally, there is no information regarding the timing of the outcome (hypertension) or the duration of follow-up for individuals (i.e., the time between the second weight assessment and the onset of hypertension or study endpoint).
Furthermore, the authors have not addressed whether any participants died during the 2011–2013 period or explained how such cases were treated in the analysis. In this context, the term "cumulative incidence of hypertension" is inappropriate. Overall, I believe the study design aligns more closely with a cross-sectional study rather than a cohort study.
2. Baseline Characteristics:
The study does not account for important baseline characteristics of the participants, particularly chronic diseases such as diabetes mellitus (including prediabetes), dyslipidemia, or oncologic conditions. These factors could significantly contribute to changes in body weight and blood pressure, and their omission limits the robustness of the findings.
Conclusion:
The study addresses an important public health issue regarding BMI and hypertension in a Japanese population, which could provide valuable insights if the methodology were improved. I believe that the authors could improve the study by specifying the follow-up duration for each participant and clarifying how missing data (e.g., deaths) were handled
Author Response
The manuscript by Kosami et al. evaluates the associations between BMI, body weight changes, and the risk of hypertension using data from a Japanese population. While the study addresses a topic of potential significance, I must highlight several methodological weaknesses and issues in the presentation of results that hinder the ability to draw clear and reliable conclusions.
Response: Thank you very much for your kind review and thoughtful summary. We have revised our manuscript according to your valuable suggestions.
1.StudyDesign:
The authors claim this is a retrospective cohort study. However, key elements of a cohort study, such as the index date, data collection period, and follow-up duration, are not clearly presented. For instance, the body weight data were collected in 2011 and 2013, but the personal interval between these two years may vary significantly (between 366 and 731 days). Additionally, there is no information regarding the timing of the outcome (hypertension) or the duration of follow-up for individuals (i.e., the time between the second weight assessment and the onset of hypertension or study endpoint).
Response: Thank you for your insightful comments regarding the clarity of our study design. This study is based on data obtained from annual medical check-ups, a unique aspect of Japan's healthcare system. These annual check-ups involve regular health assessments, including blood tests and body measurements, allowing for consistent and systematic data collection over time. In our study, we specifically targeted individuals who underwent medical check-ups in both 2011 and 2013, with a two-year interval between these assessments. In this study, the 2011 check-up served as the baseline, with baseline weight and weight changes over the two-year period evaluated as predictors of hypertension diagnosed in 2013. Therefore, this can be considered a two-point cohort study. In addition, thank you for highlighting the potential variability in intervals between the 2011 and 2013 body weight measurements. It is common for individuals to undergo annual health check-ups at approximately the same time each year. While we acknowledge the possibility of slight variations in the interval between measurements, most participants were followed over a period of approximately two years, minimizing potential bias. Additionally, blood pressure and body weight were assessed simultaneously during each health check-up, ensuring consistency in data collection. We have added this explanation to clarify the methodology and better align the text with the definition of a cohort study as following
“This is a retrospective two-point cohort study using data from the Specific Health Checkup (SHC) system, which is a nationwide annual health screening program specifically targeting metabolic syndrome. In this study, the 2011 check-up served as the baseline, with BMI and body weight changes over the two-year period (2011-2013) evaluated as predictors of hypertension development in 2013, making this a two-point cohort study.”
Response: Thank you very much for your kind suggestions. We have revised our manuscript according to your valuable suggestions.
Furthermore, the authors have not addressed whether any participants died during the 2011–2013 period or explained how such cases were treated in the analysis. In this context, the term "cumulative incidence of hypertension" is inappropriate. Overall, I believe the study design aligns more closely with a cross-sectional study rather than a cohort study.
Response: Thank you for your comment regarding the consideration of mortality during the 2011–2013 period. As this study is a two-point cohort analysis focusing on individuals who attended both the 2011 and 2013 health check-ups, cases of mortality were not included in the analysis. However, given that the study population consists primarily of individuals under 75 years of age who are generally healthy, we believe that the number of deaths during this period is likely to be very small and would have minimal impact on the study results. In response to your concern regarding terminology, we have revised the description to ensure accuracy in presenting the study design as a two-point cohort study, as follows:
Abstract: “A retrospective two-point cohort study was conducted using data from the Japanese Specific Health Checkup program from 2011 to 2013.”
Methods: “This is a retrospective two-point cohort study using data from the Specific Health Checkup (SHC) system, which is a nationwide annual health screening program specifically targeting metabolic syndrome.”
Regarding the term "cumulative incidence of hypertension," it has also been used in previous two-point cohort studies spanning five years (Hypertension 71(1):78-86.2018 PMID: 29203632; Hypertension;69(6):1036-1044.2017 PMID: 28396536). In the study, it is intended to represent the cumulative incidence of hypertension over the two-year period, which we believe is appropriate. We hope for your kind understanding on this matter.
2.Baseline Characteristics:
The study does not account for important baseline characteristics of the participants, particularly chronic diseases such as diabetes mellitus (including prediabetes), dyslipidemia, or oncologic conditions. These factors could significantly contribute to changes in body weight and blood pressure, and their omission limits the robustness of the findings.
Response: Thank you for your comment regarding the baseline characteristics of participants. In response to your suggestion, we have conducted additional analyses including diabetes mellitus and dyslipidemia, defined according to widely accepted guidelines. The results of these analyses remain consistent with our original findings. We have revised the manuscript to reflect these updated results throughout the analysis sections. As for oncologic conditions, we did not include these in the analysis, as the study population primarily consists of individuals undergoing health check-ups, who are generally healthy. We believe that the inclusion of cancer cases would have minimal impact on our findings, given the nature of this cohort.
As for oncologic conditions, we did not include these in the analysis because the study population primarily consists of individuals undergoing annual health check-ups, who are generally healthy. Additionally, given the relatively short two-year follow-up period, we believe that the inclusion of cancer cases would have minimal impact on our findings.
Conclusion:
The study addresses an important public health issue regarding BMI and hypertension in a Japanese population, which could provide valuable insights if the methodology were improved. I believe that the authors could improve the study by specifying the follow-up duration for each participant and clarifying how missing data (e.g., deaths) were handled
Response: Thank you for your valuable feedback and for recognizing the importance of this study in addressing BMI and hypertension in the Japanese population. We appreciate your suggestions and have taken steps to clarify these points in the revised manuscript.
Regarding the follow-up duration, this study utilizes data from annual health check-ups conducted in 2011 and 2013. Participants typically undergo annual health check-ups at similar times each year. While there may be slight variations in the interval between measurements, most participants were followed over a period of approximately two years, which minimizes potential bias.
For missing data (e.g., deaths), only individuals who attended both the 2011 and 2013 health check-ups were included. Given that the study population primarily consists of generally healthy individuals under 75 years of age, we believe the impact of deaths on the study findings is minimal.
We hope these clarifications address your concerns and enhance the transparency and robustness of the study methodology. Thank you for your constructive comments, which have helped us improve the manuscript.
Reviewer 2 Report
Comments and Suggestions for Authors
Koki Kosami and colleagues have studied and reported a retrospective study on the Japanese population and found an association between body weight and hypertension.
Although the overall manuscript is written well, these concerns should be addressed.
There is a selection bias. The study includes only individuals with complete health check records for 2011 and 2013. Was there a specific reason for this selection? Also, this was a decade back. Is this because of the data accessibility for those years and not more recent years? Also, retrospective design makes establishing causality between BMI/weight changes and hypertension development difficult.
The study focuses solely on hypertension as an outcome, neglecting other obesity-related comorbidities like diabetes or dyslipidemia, which could provide a more comprehensive understanding of the impact of BMI and weight changes.
Although the study adjusts for lifestyle factors (smoking, drinking, and exercise), it does not account for diet, socioeconomic status, or genetic predispositions, which could also influence both BMI and hypertension risk.
The BMI and weight change categories might oversimplify complex relationships, particularly for individuals near category thresholds.
The two-year interval between 2011 and 2013 may not capture the long-term effects of BMI and weight changes on hypertension development.
Relying on odds ratios for the entire population might obscure nuances or subgroup differences, particularly in smaller BMI categories.
While differences by sex are highlighted, hormonal factors like menopause are mentioned but not directly assessed, leaving potential biological explanations speculative.
Potential environmental or cultural contributors (e.g. stress levels) to hypertension are discussed but not quantitatively evaluated.
Author Response
There is a selection bias. The study includes only individuals with complete health check records for 2011 and 2013. Was there a specific reason for this selection? Also, this was a decade back. Is this because of the data accessibility for those years and not more recent years? Also, retrospective design makes establishing causality between BMI/weight changes and hypertension development difficult.
Response: Thank you very much for your kind review and insightful questions. The selection of individuals with complete health check records for 2011 and 2013 was due to data availability constraints. Unfortunately, more recent comprehensive data were not accessible at the time of our analysis. We acknowledge that this introduces a potential selection bias and may affect the generalizability of our findings. We have added this limitation in the Discussion section as below.
“Second, this study has a selection bias because our data were limited to 2011 and 2013 due to data availability. Moreover, this study cannot show causality between BMI/weight changes and the development of hypertension due to its retrospective observational design.”
The study focuses solely on hypertension as an outcome, neglecting other obesity-related comorbidities like diabetes or dyslipidemia, which could provide a more comprehensive understanding of the impact of BMI and weight changes.
Response: Thank you very much for your valuable suggestions. We conducted additional analyses with diabetes (defined as HbA1c≧6.5%) and dyslipidemia (LDL cholesterol ≧140 mg/dL, HDL cholesterol <40 mg/dL, or triglyceride ≧150 mg/dL) based on international and Japanese guidelines. The sensitivity analyses showed similar results. Therefore, our conclusion remains unchanged. We have added these additional results as follows.
Methods section: “In the additional sensitivity analyses, we adjusted for diabetes (defined as HbA1c≧6.5%) and dyslipidemia (LDL cholesterol ≧140 mg/dL, HDL cholesterol <40 mg/dL, or triglycerides ≧150 mg/dL) according to the International Expert Committee and the Japan Atherosclerosis Society guidelines.”
Limitation section: “To address this limitation, we conducted additional sensitivity analyses that included diabetes and dyslipidemia as covariates. The results, shown in Figure 5 and Figure 6, were similar to the main results described above.”
Although the study adjusts for lifestyle factors (smoking, drinking, and exercise), it does not account for diet, socioeconomic status, or genetic predispositions, which could also influence both BMI and hypertension risk.
Response: Thank you very much for your valuable comments. In this study, we could not obtain information on diet, socioeconomic status, or genetic predispositions. However, we conducted additional sensitivity analyses by adding diabetes and dyslipidemia as covariates to partially address this limitation. Moreover, we have included these limitations in the Discussion section as follows.
“Additionally, this study lacks some important information, such as diet, socioeconomic status, and genetic predispositions, due to the nature of a retrospective database study. Therefore, further studies are needed to incorporate this information.”
The BMI and weight change categories might oversimplify complex relationships, particularly for individuals near category thresholds.
The two-year interval between 2011 and 2013 may not capture the long-term effects of BMI and weight changes on hypertension development.
Response: Thank you very much for your important suggestions. We agree with them and have added these limitations in the Discussion section as follows. We appreciate it.
“Fifth, we created and used categories of BMI and body weight change in this study. How-ever, this method might oversimplify complex relationships, particularly for individuals near category thresholds. Moreover, this study assessed only a two-year interval, and therefore, it may not capture the long-term effects of BMI and weight changes on the development of hypertension.”
Relying on odds ratios for the entire population might obscure nuances or subgroup differences, particularly in smaller BMI categories.
Response: Thank you very much for your insightful comment. We acknowledge that relying on odds ratios for the entire population may obscure nuances or subgroup differences, particularly in smaller BMI categories. To address this, we conducted subgroup analyses stratified by BMI categories to better understand potential differences. The results were consistent with our overall findings; however, we recognize that some variations may still exist due to the limited sample size in certain subgroups. We have included this limitation in the Discussion section and noted the need for further studies with larger sample sizes to explore these subgroup differences in more detail.
“Sixth, we relied on odds ratios for the entire population in this study, which may obscure subgroup differences, particularly in smaller BMI categories. To address this, we conducted subgroup analyses by BMI categories, which yielded con-sistent results. However, variations may still exist due to the limited sample sizes, and further studies are needed to explore these differences.
While differences by sex are highlighted, hormonal factors like menopause are mentioned but not directly assessed, leaving potential biological explanations speculative.
Response: We agree with your valuable suggestions. While our study did not directly assess hormonal factors like menopause, we have strengthened our hypothesis with substantial evidence for biological mechanisms from recent research. We have added some sentences and references as follows.
“In fact, sex differences are well-documented, and recent research has directly assessed hormonal factors related to menopause (JAMA Netw Open. 2024;7(8):e2430839; BMJ 2024;387:e078784). These studies provide substantial evidence for biological explanations of biological mechanisms underlying health changes in postmenopausal women, including increased cardiovascular risk, bone density loss, and metabolic alterations. By integrating these findings, our study aligns with the broader body of evidence supporting hormonal influences on sex-specific health outcomes.”
Potential environmental or cultural contributors (e.g. stress levels) to hypertension are discussed but not quantitatively evaluated.
Response: Thank you for your insightful comment. We acknowledge that potential environmental or cultural contributors, such as stress levels, were discussed but not quantitatively evaluated in this study. Unfortunately, due to data limitations, we could not obtain quantitative measures for these factors. We have highlighted this as a limitation in the Discussion section and noted that future studies incorporating quantitative evaluations of these contributors are warranted to provide a more comprehensive understanding of hypertension risk.
“Seventh, although potential environmental or cultural contributors, such as stress levels, are important factors for hypertension, we could not assess them due to data limitations. This remains a limitation, and future studies incorporating these factors are needed for a more comprehensive understanding of hypertension risk. “
Reviewer 3 Report
Comments and Suggestions for Authors
The article addresses a highly relevant topic: the utility of Body Mass Index and Weight Change as predictors of the development of hypertension. This is examined using a longitudinal study conducted on a large Japanese population, which lends the results significant statistical power.
The introduction, despite citing numerous references, is too brief and does not adequately frame the study's topic. Additionally, many of the references are outdated, as indicated by the high obsolescence index (median age of 8 years). It would be advisable to include newer references or replace the older ones with more recent studies.
The study design is excellent, well-explained, and capable of achieving the proposed objectives. However, instead of the current flowchart, we believe it would be more appropriate to use a PRISMA model flowchart. The inclusion/exclusion criteria are missing and should be incorporated. Additionally, a statistical analysis section should be added.
The results, although simple, are very interesting and are appropriately presented, supported by tables and figures that enhance their readability and comprehension.
The discussion is adequate but, like the rest of the text, relies on outdated references (overall obsolescence index of 9 years). At the end of the discussion, a paragraph outlining the study's strengths and weaknesses should be added.
The conclusions are concise, as is appropriate for this section, and they adequately address the proposed objectives.
Recommendations:
- Update the article's general bibliography to reduce the obsolescence index.
- Modify the flowchart to follow the PRISMA model.
- Include inclusion/exclusion criteria.
- Add a statistical analysis section.
- Add a section discussing strengths and weaknesses.
Author Response
The article addresses a highly relevant topic: the utility of Body Mass Index and Weight Change as predictors of the development of hypertension. This is examined using a longitudinal study conducted on a large Japanese population, which lends the results significant statistical power.
Response: Thank you very much for your kind review and thoughtful summary.
The introduction, despite citing numerous references, is too brief and does not adequately frame the study's topic. Additionally, many of the references are outdated, as indicated by the high obsolescence index (median age of 8 years). It would be advisable to include newer references or replace the older ones with more recent studies.
Response: Thank you very much for your suggestions. We have replaced some older references with more recent and updated studies to improve the relevance and quality of the introduction.
The study design is excellent, well-explained, and capable of achieving the proposed objectives. However, instead of the current flowchart, we believe it would be more appropriate to use a PRISMA model flowchart. The inclusion/exclusion criteria are missing and should be incorporated. Additionally, a statistical analysis section should be added.
Response: Thank you very much for your kind comments and valuable suggestions. We have revised the flowchart according to the PRISMA model and included the inclusion/exclusion criteria in the 2.2 Inclusion and Exclusion Criteria section. Furthermore, we have included the number of participants for each criterion in the 3.1 Participants section, not in the 3.4. statistical analysis section.
The results, although simple, are very interesting and are appropriately presented, supported by tables and figures that enhance their readability and comprehension.
Response: Thank you very much for your kind comments and thoughtful review.
The discussion is adequate but, like the rest of the text, relies on outdated references (overall obsolescence index of 9 years). At the end of the discussion, a paragraph outlining the study's strengths and weaknesses should be added.
Response: Thank you very much for your kind comments. We have replaced some older references with more recent studies to improve the relevance and quality of the discussion. We have also added a paragraph outlining the study's strengths and weaknesses at the end of the discussion as shown below.
“Strengths and Weaknesses of the Study
This study's strengths include a large sample size of 397,181 participants, providing robust statistical power, and the use of standardized data from the Japanese Specific Health Checkup program. The analysis of sex differences offers valuable insights into hypertension risk factors for men and women.
However, the two-year interval may not capture long-term effects, and factors such as diet, socioeconomic status, and stress levels were not evaluated due to data limitations. Additionally, categorizing BMI and weight changes may oversimplify complex relationships, particularly near category thresholds. Furthermore, as a retrospective cohort study, it cannot establish causal relationships.”
The conclusions are concise, as is appropriate for this section, and they adequately address the proposed objectives.
Response: Thank you very much for your kind comments. We sincerely appreciate your thoughtful review.
Recommendations:
Update the article's general bibliography to reduce the obsolescence index.
Modify the flowchart to follow the PRISMA model.
Include inclusion/exclusion criteria.
Add a statistical analysis section.
Add a section discussing strengths and weaknesses.
Response: Thank you very much for your summarized recommendations. We have revised everything according to your kind suggestions.
Round 2
Reviewer 1 Report
Comments and Suggestions for Authors
No additional comments. Thanks
Reviewer 3 Report
Comments and Suggestions for Authors
Is ok now